# Chronic Supplementation of 2S-Hesperidin Improves Acid-Base Status and Decreases Lactate at FatMax, at Ventilatory Threshold 1 and 2 and after an Incremental Test in Amateur Cyclists

**DOI:** 10.3390/biology11050736

**Published:** 2022-05-11

**Authors:** Francisco Javier Martínez-Noguera, Pedro E. Alcaraz, Jorge Carlos-Vivas, Cristian Marín-Pagán

**Affiliations:** 1Research Center for High Performance Sport, Catholic University of Murcia (UCAM), Campus de los Jerónimos Nº 135, 30107 Murcia, Spain; palcaraz@ucam.edu (P.E.A.); cmarin@ucam.edu (C.M.-P.); 2Health, Economy, Motricity and Education Research Group (HEME), Faculty of Sport Sciences, University of Extremadura, Avda. de Elvas, s/n., 06006 Badajoz, Spain; jorge.carlosvivas@gmail.com

**Keywords:** exercise, endurance, performance, sport nutrition, polyphenols, flavonoids, metabolism

## Abstract

**Simple Summary:**

Currently, hesperidin is a molecule found mainly in citrus fruits and is being widely researched in the area of chronic disease, but also in the field of sports nutrition. Some studies have shown its antioxidant, anti-inflammatory, lipid and carbohydrate metabolism modulating effects, including the enhancement of nitric oxide synthesis. However, few human studies have demonstrated a positive effect of hesperidin intake, in particular 2S-hesperidin, on sports performance, particularly in anaerobic and aerobic tests. However, the biochemical mechanisms that may be responsible for this enhanced performance have not yet been described. Therefore, one of the aims of this study was to assess whether an eight-week intake of 2S-hesperidin can improve acid-base status and metabolic status (lactate and glucose) in an incremental test in amateur cyclists. The results showed that amateur cyclists chronically supplemented with 2S-hesperidin improved acid-base status and lactate at FatMax, ventilatory thresholds 1 and 2, and in the acute phase of recovery after maximal effort.

**Abstract:**

Chronic supplementation with 2S-hesperidin improves performance; however, the mechanisms underlying this effect have not yet been explored. Therefore, the aim of this study was to assess whether changes in acid-base status may be associated with improved performance after 2S-hesperidin supplementation compared to microcellulose (placebo). Forty amateur cyclists (n = 20 per group) underwent a rectangular test where capillary blood samples were taken at baseline, FatMax1, VT1, VT2, P_MAX_, FatMax2 and EPOC to measure acid-base parameters. After eight weeks of 2S-hesperidin supplementation (500 mg/d) increased HCO_3_^−^, SBC, ABE (*p* ≤ 0.05) and decreased Lac were found at FatMax1, VT1, FatMax2 and EPOC (*p* ≤ 0.05), while decreased Lac at VT2 was found with a large effect size (ES = 1.15) compared to placebo. Significant group differences in the area under the curve were observed when comparing pre-post-intervention pH changes (*p* = 0.02) between groups. Chronic supplementation with 2S-hesperidin improved acid-base status and Lac, both at low-moderate and submaximal intensities, improving recovery after exercise-to-exhaustion in amateur cyclists.

## 1. Introduction

Acid-base state and lactate (Lac) are metabolic components that can be altered depending on the exercise intensity level because of changes in blood gases (O_2_ and CO_2_) and metabolic markers (metabolites derived from the metabolism of the phosphocreatine and glycolysis) [1]. Moreover, nutritional composition, such as in the ketogenic diet, can modify the acid-base state. [2].

The main components that modulate the acid-base state are pH, bicarbonate (HCO_3_^−^) and carbon dioxide (CO_2_) concentrations. Increased aerobic metabolism affects pH by increasing the production of CO_2_ and HCO_3_^−^, while high intensity exercise promotes the accumulation of anaerobic end products, particularly lactic acid [3]. This metabolic environment leads to muscle and blood acidification (↓pH) which can cause enzymatic and metabolic impairment and fatigue anticipation [4]. Some authors have shown how intracellular acidification decreases oxidative phosphorylation, and possibly reduces mitochondrial oxidative capacity, in isolated fibers [5], in muscle [6], as well as in in vivo muscle [7]. Therefore, the accumulation of protons (H^+^) lowers pH at high exercise intensity levels, inhibits oxidative ATP supply (i.e., affecting mitochondrial capacity), and leads to an inability to maintain VO_2MAX_, which is an important factor that limits sustained oxygen consumption and performance [4].

There are several factors, mainly training, nutrition and ergogenic aids, that affect endurance performance [8]. Endurance training produces a number of adaptations at cellular and systemic levels with the aim of reducing the breakdown of the whole-body homeostasis caused by exercise [9,10]. Type of training, training status, as well as diet composition affect exercise response and adaptations in a positive or negative way [11,12]. In addition, the intake of ergogenic aids modulate the adaptations generated by resistance training, like in reactive oxygen species (ROS) signaling, acid-base balance, the redox state, training load, etc. [11]. These physiological mechanisms allow the body to adapt to training and improve the athlete’s performance.

Endurance athletes are increasingly using ergogenic aids to improve performance, and these include dietary nitrates [13], β-alanine [14], antioxidants [15], sodium bicarbonate [16], creatine [17] and polyphenols [18]. There is a recent worldwide research interest in the pleiotropic effect of polyphenols on the immune system, chronic diseases, and aging [19,20,21,22,23]. Polyphenols are a large family of compounds divided into four groups: flavonoids (e.g., hesperidin, hesperetin, etc.), phenolic acids, stilbenes and lignans [19].

Specifically, hesperidin is a flavanone flavonoid, being the predominant one in some citrus fruits [24] such as sweet orange (*Citrus sinensis*) [25]. Hesperidin may be found as 2S or 2R isomers, but 2S-hesperidin is the natural predominant form in citrus [26]. During conventional extraction procedures hesperidin undergoes a transformation from an S to an R isomer [27]. Hesperidin has been shown to improve performance and the endogenous antioxidant system [28,29,30,31]. A six-week consumption of hesperetin (50 mg·kg^−1^·d^−1^), the main metabolite of hesperidin, has been shown to improve antioxidant status (reduced glutathione (GSH), oxidized glutathione (GSSG) and GSH/GSSG ratio) and running performance (exercise time) in aged mice [28]. Similar results were found with a five-week intake of 2S-hesperidin (200 mg/kg), where improvements in the test performance (running until exhaustion) (+58%) and in the antioxidant system (superoxide dismutase (SOD), glutathione peroxidase (GPx)) in the liver and lymphoid tissue were observed in rats [29]. In addition, Martínez-Noguera et al. [31] showed improvements in average power (+2.3%), maximum speed (+3.2%) and total energy (∑ 4 sprint test) (+2.6%) during a repeated sprint test (4 sprints of 30 s) following an acute intake of 2S-hesperidin (500 mg) in amateur cyclists. Recently, the same authors demonstrated significant performance improvements in estimated functional threshold power (eFTP) (+2.3%) and maximum power (+1.9%) during an incremental test after eight-weeks of 2S-hesperidin (500 mg/d) consumption in amateur cyclists [30]. An increase in power at maximum speed (+1.1%) and a decrease in time at peak power (−11.2%) in the Wingate test (one sprint of 30 s) was also observed [30].

To our knowledge, the effect of 2S-hesperidin intake on acid-base state and lactate (Lac) has not been yet studied. A matrix of Okra-derived polyphenols (mainly flavonoids, isoquercetin, and quercetin-3-O-gentiobiose) and lemon seeds have been shown to decrease lactate after exercise-to-exhaustion in animals [32,33]. In addition, at the molecular level, hesperetin (a metabolite of hesperidin) has shown an increase in the activation of the nuclear respiratory factors (NRF) and peroxisome proliferator-activated receptor-gamma coactivator 1α (PGC-1α) [28], which increases mitochondrial biogenesis [34]. We hypothesized that chronic 2S-hesperidin supplementation decreases lactate production through mitochondrial enhancement mediated by an increase in the activation of NRF and PGC-1α.

The main objective of this intervention study was to determine whether supplementation with 500 mg/d 2S-hesperidin for eight weeks improves markers of acid-base status and decreases lactate in amateur cyclists, based on a capillary blood sample at baseline, during and after the rectangular test in amateur cyclists.

## 2. Methodology

### 2.1. Participants

Forty healthy male amateur cyclists completed the study (Table 1). The inclusion criteria were as follows: aged 18–55 years, BMI 19–25.5 kg m^−2^, have at least three years of cycling experience and undergo 6–12 h·week^−1^ of training. Participants were excluded if they: (a) were smokers or regular alcohol drinkers, (b) had metabolic, cardiorespiratory or digestive pathologies or anomalies, (c) had an injury in the last six months, (d) were consuming any type of supplementation or drug in the prior two weeks and (e) had no normal values in some parameter of the baseline health blood analysis. Eligible participants gave and signed informed consent prior to the start of the study. The study was conducted according to the guidelines of the Helsinki Declaration for Human Research [35] and was approved by the Ethics Committee of the Catholic University of Murcia (CE091802), registered in ClinicalTrials.gov (Identifier: NCT04597983). All participants completed the study.

### 2.2. Study Design

This research was part of a larger, previously published study that investigated the effect of chronic hesperidin intake on performance [30] and body composition [36] in amateur cyclists. To carry out this study, a double-blind, parallel and randomized experimental design was conducted. Using Randomizer software [37], participants were randomized into two groups: experimental (2S-hesperidin; n = 20) and control (Placebo; n = 20). Participants either took two capsules of 2S-hesperidin (500 mg in total/d of 2S-hesperidin; HealthTech BioActives, Murcia, Spain) or placebo (500 mg of microcellulose/d) for eight weeks. Cardiose was the product used in this study containing 2S-hesperidin (NLT 85% 2S-hesperidin). The cyclists continued their normal training schedule during the length of the study. Both groups had similar general and training characteristics at the start of the study (Table 1 and Table 2). Subjects in both groups were instructed not to consume foods high in citrus flavonoids (grapefruit, lemons, or oranges) for five days prior to and during the study, which was verified by diet recall questionnaires.

### 2.3. Procedures

The experimental protocol required five visits to the laboratory. Day 1 consisted of a medical examination and blood analysis to determine overall health status. Days 2 and 4 entailed a 24-h diet recall questionnaire and an incremental test until exhaustion on a cycle ergometer. Days 3 and 5 involved a 24-h diet recall and a rectangular test on a cycle ergometer (Figure 1 and Table 3). Participants consumed a standardized breakfast 2.5 h before each testing session (visits 2, 3, 4 and 5) that consisted of 95.2 gr carbohydrates (68%), 18.9 gr protein (14%) and 11.3 gr lipids (18%), which was prescribed by a sports nutritionist.

### 2.4. Testing

#### 2.4.1. Medical Exam

The medical exam consisted of a health history, resting electrocardiogram and examination (auscultation, blood pressure, etc.) by a medical doctor to confirm that the cyclist was healthy enough to participate in the study.

#### 2.4.2. Blood Samples

Venous blood extraction was conducted by a certified nurse, where one 3-mL tube of ethylenediaminetetraacetic acid (EDTA) for hemogram and another 3.5-mL tube with polyethene terephthalate (PET) for health analysis were obtained. A red blood cell count was carried out in an automated Cell-Dyn 3700 analyzer (Abbott Diagnostics, Chicago, IL, USA), using internal (Cell-Dyn 22) and external (Program of Excellence for Medical Laboratories-PEML) controls. Values of erythrocytes, hemoglobin, hematocrit and hematimetric indexes were estimated.

#### 2.4.3. Maximal Test

An incremental step with a final ramp test using a metabolic cart (Metalyzer 3B. Leipzig, Germany) was performed to determine the maximal zone of fat-burning (FatMax), ventilatory thresholds 1 (VT1) and 2 (VT2), maximum power output (P_MAX_) and maximal oxygen uptake (VO_2MAX_). The test started at 35W and was increased by 35W every 2 min, followed by a final ramp (+35W·min^−1^), which started when the RER was higher than 1.05, to exhaustion. To guarantee that VO_2MAX_ was obtained, the following criteria were checked: plateau in the final oxygen consumption (VO_2_) values (increase ≤2.0 mL·kg^−1^·min^−1^ in the last 2 loads), maximal theoretical heart rate (HR) (220-age)·0.95) for a cycling test [38], RER ≥ 1.15 and lactate ≥8.0 mmol·L^−1^ [39,40]. Ventilatory thresholds were obtained using the ventilatory equivalents method described by Wasserman [41].

#### 2.4.4. Rectangular Test

The rectangular test was performed on a cycle ergometer using the power output values obtained from the maximal test (FatMax, VT1, VT2 and P_MAX_). The test consisted of 10-min at FatMax1 (intensity at which maximum fat oxidation is given), 10-min at VT1 and 10-min at VT2, until exhaustion at P_MAX_ and 15-min at FatMax2. As soon as the exercise was completed, the subjects immediately laid down on a bed so that excess post-exercise oxygen consumption (EPOC) could be measured for 30 min. Cardiorespiratory variables (VO_2_, VO_2_R, carbohydrate oxidation (CHO), fat oxidation (FAT) and cycling economy) were determined for the different metabolic zones. These metabolic data have been published separately [30].

#### 2.4.5. ABL-90 (Blood Gas Analyzer)

The acid-base status and metabolic biomarkers (pH, bicarbonate (HCO_3_^−^), standard bicarbonate (SBC), actual bass excess (ABE), standard bass excess (SBE) lactate (Lac) (amperometry electrode using lactate oxidase) and glucose (Glu)) were determined from arterialized capillary blood at the fingertip at rest (pre), in the last 30 s of FatMax1, in the last 30 s of VT1, in the last 30 s of VT2, Post P_MAX_, in the last 30 s of FatMax2 and at the end of EPOC (at rest). The ABL 90 FLEX blood gas analyzer (Radiometer Medical ApS, Copenhagen, Denmark) was used to measure the above-mentioned parameters and was calibrated at hourly intervals throughout the day, with internal reference standards. A previous study showed that ABL90 FLEX had good accuracy [42]. Plastic capillary tubes were pre-heparinized with electrolytically balanced solid heparin, significantly reducing the risk of clots and helping to ensure reliable results without electrolyte bias.

### 2.5. Statistical Analyses

Data analysis was conducted using IBM Social Sciences software (SPSS, version 21.0, Chicago, IL, USA). Descriptive statistics are presented as mean and standard deviation (SD). Levene’s and Shapiro-Wilk tests checked the homogeneity and normality of the data, respectively. A group × time × moment ANOVA analyzed within-group and between-group differences in all dependent variables and for every time-point of measurement (baseline (pre), FatMax1, VT1, VT2, P_MAX_, FatMax2 y EPOC) and in both moments (pre-test and post-test). In addition, the area under the curve (AUC), resulting from the integration of the three time-points of measurement taken during the rectangular test was calculated for each variable. The AUC was used to analyze pre-post differences both within groups and between groups. A repeated-measures t-test analyzed within-group differences in the AUC, and between-group comparisons in the AUC were conducted using an independent T-test. Cohen’s d effect size (ES) (95% confidence interval) was calculated for all comparisons. Threshold values for ES statistics were as follows: 0.2–0.5 small, 0.5–0.8 moderate, and >0.8 large [43]. Significant differences were considered when *p* ≤ 0.05.

## 3. Results

### 3.1. Rectangular Test

#### 3.1.1. Acid-Base Status Biomarkers (Capillary Blood Gases)

Table 3 shows the intragroup changes in biomarkers of the acid-base status at baseline, FatMax1, VT1, VT2, P_MAX_, FatMax2 and EPOC achieved during the rectangular test. No significant changes post-intervention in 2S-hesperidin group were found. Similarly, no significant changes were found at baseline post-intervention (Table 4 and Figure 2) in placebo.

When we measured acid-base status at FatMax1 (Table 4 and Figure 2), the 2S-hesperidin group showed a significant increase in HCO_3_^−^ (10.30%; *p* = 0.040; ES = 4.85), and SBE (424%; *p* = 0.046; ES = 4.87).

When we analyzed the changes in VT1 (Table 4 and Figure 2), 2S-hesperidin supplementation resulted in a significant increase in HCO_3_^−^ (5.55%; *p* ≤ 0.001; ES = 3.20), SBC (4.65%; *p* = 0.001; ES = 3.59), ABE (6500%; *p* = 0.001; ES = 3.48) and SBE (1913%; *p* = 0.001, ES = 3.42). A significant difference in SBE (*p* = 0.01; ES; 4.59) and in ΔVT1 was observed between groups (Table 5).

During the rectangular test, after P_MAX_ decreased in intensity to FatMax2 (Table 4 and Figure 2), 2S-hesperidin showed a significant increase in pH (0.26%; *p* = 0.028; ES = 1.48), HCO_3_^−^ (10.81%; *p* = 0.003; ES = 2.19), SBC (7.68%; *p* = 0.006; ES = 2.01) and ABE (34.23%; *p* = 0.034; ES = 1.62).

Finally, at EPOC (Table 4 and Figure 2), the 2S-hesperidin group showed a significant increase in HCO_3_^−^ (5.76%; *p* = 0.001; ES = 2.63), SBC (4.48%; *p* = 0.001; ES = 2.39), ABE (246.15%; *p* = 0.001; ES = 2.42) and SBE (248.39%; *p* ≤ 0.001; ES = 2.53). In the placebo group (Table 4), we found a significant increase in HCO_3_^−^ (3.40%; *p* = 0.045; ES = 1.36) at resting EPOC after intervention.

When comparing the intra-group AUCs of the acid-base and metabolic biomarkers, there was a significant increase in pH (0.16%; *p* = 0.016; ES = 0.54), HCO_3_^−^ (6.34%; *p* = 0.012; ES = 0.74) SBC (5.07%; *p* = 0.017; ES = 0.79). After comparing AUCs in cyclists having placebo, there was no significant change after the intervention (Table 4). There were significant group differences in ∆AUC in pH (*p* = 0.02; ES = 1.03), where post-intervention was found in favor of 2S-hesperidin supplementation (Table 5).

#### 3.1.2. Metabolic Biomarker (Capillary Blood)

When we measured Lac at FatMax1 (Table 4 and Figure 2), the 2S-hesperidin group showed a significant decrease in Lac (−29.39%; *p* = 0.010; ES = 2.26) post-intervention. In addition, in the placebo group, we found a significant decrease in Lac (−21.91%; *p* = 0.041; ES = 1.71) post-intervention (Table 4). On the other hand, we also observed a significant decrease in Lac (−30.83%; *p* = 0.003; ES = 2.98) at VT1 (Table 4 and Figure 2) after the supplementation period. However, no significant changes were found in the placebo group post-intervention.

In the submaximal exercise stage measurements (VT2) (Table 4 and Figure 2), the 2S-hesperidin group did not show significant changes in the acid-base state post-intervention, but a large effect size (*p* = 0.13; ES; 1.49) was observed in VT2. Similarly, there were no significant changes in VT2 in placebo after the eight-week intervention. In line with the previous results, we found a significant decrease at FatMax2 (−18.56; *p* = 0.018; ES = 1.88) and at EPOC (−18.56%; *p* = 0.039; ES = 1.51) in Lac post-intervention after 2S-hesperidin supplementation (Table 4 and Figure 2). There was no change in the placebo group.

When comparing the intra-group AUC, we showed a downward trend in Lac (12.58%; *p* = 0.057; ES = 0.51) in 2S-hesperidin post-intervention

## 4. Discussion

This study primarily aimed to determine if eight-weeks of 2S-hesperidin supplementation (500 mg/d) was able to modify biomarkers of the acid-base state (during incremental test and recovery), and whether these possible changes can affect performance in trained amateur cyclists. The most important findings of this study showed that the chronic intake of 2S-hesperidin increased acid-base status markers and reduced Lac at FatMax1, VT1, FatMax2 and EPOC compared to placebo. In addition, a downward trend was observed with a large effect size at VT2 in the 2S-hesperidin group. However, there were no pre-post changes in the aforementioned parameters in the placebo group. On the other hand, significant differences were also found when comparing AUC between groups for pH.

### 4.1. Changes in Acid-Base Status at Baseline

Between the end of September and December (when our study was conducted), cyclists decreased the volume and intensity of training, which leads to detraining. This is defined as the partial or complete loss of physiological, anatomical and performance adaptations due to the reduction or cessation of training [44]. However, in our study, we found no significant changes after the intervention in markers of acid-base status and Lac in both groups at baseline.

### 4.2. Changes in Acid-Base Markers at FatMax1 and VT1

Exercise performed in FatMax1 (intensity that produces maximum fat oxidation rate, translated as the % VO_2MAX_,) uses the oxidation of free fatty acids and intramuscular triglycerides in skeletal muscle as its main source of energy [45]. Both peak oxidative maximum and FatMax, as important determinants of performance in elite endurance athletes, are related to better prolonged exercise endurance [46]. On the other hand, being a zone close to the physiological-metabolic level of the FatMax, VT1 represents the first increase in minute ventilation (VE) that is proportional to the increase in CO_2_ output (VCO_2_) generated by the HCO_3_^−^ buffering of lactic acid. In these two exercise zones (FatMax and VT1), energy production is predominantly aerobic.

After the intervention, we showed an increase in HCO_3_^−^ in FatMax1 and VT1 in the 2S-hesperidin group. In addition, we observed an increase in SBC, ABE and SBE, with a decrease in Lac, indicating an improvement in acid-base status. These results could explain how the eight-week intake of 2S-hesperidin was able to maintain fatty acid and carbohydrate oxidation compared to placebo (2S-hesperidin: −12.9% and −0.5% vs placebo: −34.3% and 17.7%; respectively) at VT1 intensities in amateur cyclists in a period where there was lower intensity and a lower volume of training [30]. This previous study was obtained simultaneously with the current study in our laboratory. Sumi et al. [47] observed that by modifying the acid-base state (↓pH, HCO_3_^−^, ABE and ↑Lac) via hypoxia (fraction of inspired oxygen of 14.5%) after performing a high-intensity interval-type endurance exercise there was an increase in carbohydrate oxidation. Moreover, an inverse relationship between the concentration of free fatty acids and Lac in blood and plasma and oxidation has been reported [48] via the inhibition of lipolysis (Lac) in fat cells through an activation of an orphan G protein-coupled receptor (GPR81) [49]. Therefore, modifications of the acid-base state and Lac can modulate the oxidation of metabolic substrates during exercise.

It is well known that Lac is characterized by being a dynamic fuel which is produced even when there is no lack of oxygen, thereby offering a metabolic benefit [50]. Furthermore, there is strong evidence that the catabolism of glucose and glycogen leads to lactate production under fully aerobic conditions [51]. Recently, Brooks [52] proposed that improved mitochondrial mass in oxidative fibers could be implicated in the improvement of lactate metabolization (↓ levels) at the mitochondrial level. These results are in line with those found by Biesemann [28], who observed that exposure to hesperetin (hesperidin metabolite) in muscle cells increases the expression of PGC-1α and NFR-2, which are two key transcription factors in mitochondrial biogenesis.

Therefore, chronic ingestion of 2S-hesperidin could improve acid-base status and Lac, which could influence the efficiency of energy substrate utilization (↑ fats and ↓glucose oxidation) at low-to-moderate intensities (FatMax1 and VT1), leading to the conservation of muscle glycogen for the higher intensity phases in the final phase of a competition in amateur cyclists.

### 4.3. Changes in Acid-Base Markers at VT2

VT2 is an exercise zone of elevated but stable metabolic acidosis, and when this zone is exceeded, there is a fatigue-inducing accumulation of metabolites (increased lactic acid and H^+^) [53] and changes in the recruitment pattern of motor units that activate the muscle [54]. During a submaximal steady state exercise, Lac production (input) is equal to Lac clearance (output), where the concentration of Lac in the lactate pool remains constant [55]. At exercise intensities above the steady state, an increase in concentration could be attributed to an increase in the rate of lactate production or a decrease in the rate of lactate clearance [55].

In this clinical trial we observed a non-significant intra-group increase in both treatments but with a large effect size (ES = 1.07) in pH for the 2S-hesperidin group and no significant changes but a moderate effect size (ES = 0.52) (between-group comparison) in ΔVT2. Together with these changes, we also found a non-significant intra-group decrease in both groups, but with a large effect size (ES = 1.49) in 2S-hesperidin (2S-hesperidin: −16.6% vs. placebo: −3.8%) and no significant changes but a large effect size (ES = 1.15) in ΔVT2. In relation to these results, Martínez-Noguera et al. [30] found that chronic supplementation of 500 mg/d of 2S-hesperidin improved the performance of the estimated functional threshold power (eFTP) (2.3%) in an incremental test, which could be linked to lower levels of Lac in VT2 post-intervention in this study. There is a strong correlation between power output at 4 mmol·L^−1^ blood lactate and FTP in trained male cyclists, it being an exercise area near VT2 [56]. FTP is an important factor in cycling performance, as the maximum power sustained for one hour (this work zone) predicts time trial performance in elite cyclists [56]. There are currently no other studies with which the described results can be compared, as most research has used maximal intensity exercise to test polyphenol or flavonoid intake effects.

Flavonoids, such as hesperidin, can facilitate an increase in mitochondrial Ca^2+^ levels by acting on the mitochondrial Ca^2+^ uniport [57]. This mechanism can up-regulate the respiratory rate and ATP production and stimulate endothelial nitric oxide synthase (eNOS), thus increasing nitric oxide (NO) synthesis [58,59,60]. NO-induced vasodilation may increase oxygen supply to active muscles, which could improve performance [59,61]. Hesperetin also contributes to mitochondrial biogenesis [28], as mentioned above. Hypothetically, the combination of these two hesperidin induced effects (↑NO and mitochondrial biogenesis) could decrease Lac levels in VT2. Further research is needed to clarify this effect.

However, the lower lactate production in the 2S-hesperidin group seems to indicate a lower anaerobic contribution of glycolysis or a better lactate clearance, thereby leading to better pH levels. This finding could help endurance athletes, in particular cyclists, to not only prevent an increase but also to decrease Lac in submaximal exercise zones, delaying the onset of negative effects due to Lac accumulation.

### 4.4. Changes in Acid-Base Markers at FatMax2

In the rectangular test, after the maximum exercise zone, the cyclists performed a FatMax2 exercise. Only the 2S-hesperidin supplementation (as compared to placebo) led to enhanced acid-base status markers (↑pH, HCO_3_^−^, SBC and ABE) and decreased Lac in FatMax2 post-intervention. This suggests that there was a lower anaerobic contribution, as Lac is a sensitive biomarker of non-oxidative glycolysis [62], and it also indicates a better washout of Lac in the transition from post P_MAX_ (very high intensity exercise) to FatMax2 (low-moderate intensity exercise). The decrease in Lac could be explained by improvements in the intracellular to extracellular transfer of lactate for oxidation or transformation to glucose and glycogen [63,64]. Currently, we are the first to use a rectangular test. Most studies have measured Lac after exercise to exhaustion, but not after low-moderate intensity exercise (FatMax2). This makes it difficult to compare our results with other studies.

With regard to other studies using different polyphenols, Peng et al. studied the effects of purified chestnut flower flavonoids supplementation in mice and observed a decrease in Lac and an increase in lactate dehydrogenase (LDH) after a swim to exhaustion test [65]. LDH is expressed in several tissue cells and is involved in the glycolytic pathway by facilitating the redox reaction between pyruvic acid and lactic acid (reversible reaction) with the concomitant actions of NADH and NAD^+^ playing an important role in the quenching of Lac during high endurance exercise [66]. However, more research is needed to precisely explain the mechanisms by which flavonoids can decrease Lac after high-intensity exertion.

Another mechanism that may have elicited the described post-intervention decrease in Lac in FatMax2 is the increased peripheral blood flow in leg muscles via the increased production of NO. This increased flow would lead to the transport of lactate to other tissues for oxidation, allowing greater phosphocreatine resynthesis and less accumulation of metabolites, consequently enhancing recovery [67]. Therefore, based on our findings, the chronic intake of 2S-hesperidin can decrease Lac levels and improve markers of acid-base status, which offers cyclists greater recovery after maximal exertion.

These findings suggest that the experimental group (2S-hesperidin) showed less muscle fatigue and better response to training, taking into account that the study was conducted in a training period where the volume and intensity of training was reduced (end of September–mid December). The chronic intake of 2S-hesperidin would therefore improve recovery after maximal exercise followed by low-intensity exercise (FatMax2) via a decrease in Lac and acid-base status.

### 4.5. Changes in Acid-Base Markers at EPOC

During high intensity exercise of short duration (high anaerobic energy demand), there is an increase in H^+^ and a reduction in blood and muscle pH [68,69]. Elevation of blood lactate reflects that aerobic ATP generation is insufficient for the required ATP demands and needs to be supplemented by anaerobic ATP generation [68,70].

During resting EPOC, 2S-hesperidin reported an increase in markers of acid-base status (↑pH, HCO_3_^−^, SBC, ABE and SBE) and a decrease in Lac. However, in the placebo group, only an increase in HCO_3_^−^ post-intervention was detected. Currently, there are no studies that have tested the effect of polyphenols or flavonoids on acid-base status markers after a rectangular resting test. Similar results were obtained after acute ingestion of sodium bicarbonate (0.3 g·kg^−1^ body mass), which improved recovery after exercise at 100% peak power to exhaustion due to the increase in pH and HCO_3_^−^, which allowed for improved performance in a second test to exhaustion in recreationally active subjects [71].

To the best of our knowledge, this is the first time that it has been shown that flavonoid supplementation (namely 2S-hesperidin) is able to improve the acid-base status (buffering effect) at moderate-submaximal intensities and after maximal exercise. In general, the changes observed in resting EPOC are in line with those found in FatMax2, where the 2S-hesperidin group improved the acid-base status and decreased Lac post-intervention, resulting in a better recovery effect after high-intensity effort followed by low-intensity exertion when compared to placebo. In addition, reduced Lac levels in the 2S-hesperidin group indicate a lower energy contribution from carbohydrates via the anaerobic pathway. Changes produced by 2S-hesperidin intake could be important for athletes who only have a short period between competitive or training events (i.e., high intensity efforts with little recovery time).

### 4.6. Changes in Acid-Base Markers in AUCs

When assessing the AUCs, which reflect the overall changes, the 2S-hesperidin group showed an increase in capillary blood markers in pH, SBC, and HCO_3_^−^, but a decrease in Lac. In contrast, no significant changes in AUC were found in the placebo group. In addition, significant changes in pH were found in ΔAUC (*p* = 0.02; ES = 1.03) when comparing the groups.

Overall, our data confirm improvements in acid-base status (low-moderate-submaximal intensity exercise) and a decrease in Lac (low-moderate and high intensity exercise) after eight weeks of 2S-hesperidin (500 mg/d) supplementation in amateur cyclists.

## 5. Limitations

As already mentioned, the period of the season in which the study was conducted could have affected the overall results and, consequently, the interpretation of the findings. A larger sample size could have benefited the results of the study, but due to budget constraints and the planning involved to run the tests in two months, it was unfeasible to recruit more subjects.

## 6. Conclusions

Chronic supplementation of 2S-hesperidin improved acid-base status (↑pH, HCO_3_^−^ and SBC) at low-moderate exercise intensities and in recovery, and decreased Lac at low-moderate and submaximal intensities and in recovery in amateur cyclists. These findings position 2S-hesperidin as a new ergogenic aid which may help cyclists to improve performance at high intensities (VT2) and aid in recovery after very high intensity exercise (P_MAX_). Furthermore, the improvements in acid-base status and the decrease in Lac in FatMax1 and VT1 generated after 2S-hesperidin ingestion are linked to the maintenance of fatty acid oxidation in FatMax1 and VT1, indicating a greater contribution of the aerobic pathway at low-moderate intensity exercise and recovery relative to placebo.

## Figures and Tables

**Figure 1 biology-11-00736-f001:**
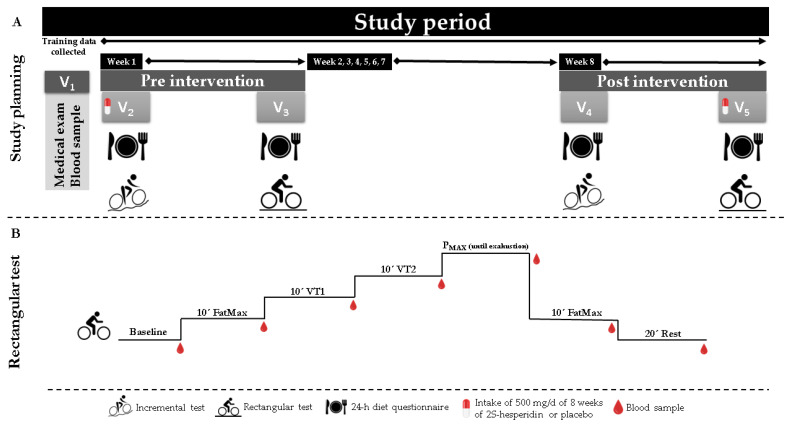
(**A**)-Study plan, 2S-Hesperidin intake was consume for eight weeks (visit two to visit five). (**B**)–Rectangular test, seven finger capillary blood samples were taken and analyzed on the ABL-90 FLEX blood gas meter.

**Figure 2 biology-11-00736-f002:**
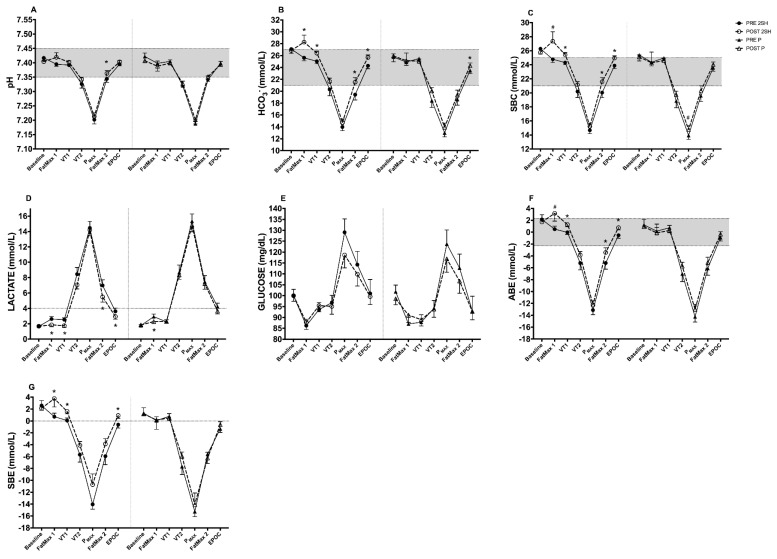
Differences between pre- and post-intervention intra-group differences in acid-base status parameters, lactate and glucose at different points of the rectangular test (**A**–**G**). * *p* < 0.05. # *p* = 0.05–0.08.

**Table 1 biology-11-00736-t001:** Baseline general characteristics and training variables of the cyclists.

	2S-Hesperidin	Placebo	*p*-Value
Age (years)	35.0 (9.20)	32.6 (8.90)	0.407
Body mass (kg)	71.0 (6.98)	70.4 (6.06)	0.773
Height (cm)	175.3 (6.20)	176.5 (6.10)	0.541
BMI (kg·m^−2^)	23.1 (1.53)	22.6 (1.43)	0.292
BF (%)	8.9 (1.63)	9.0 (1.64)	0.803
VO_2MAX_ (L·min^−1^)	3.99 (0.36)	3.98 (0.63)	0.971
VO_2MAX_ (mL·kg^−1^·min^−1^)	57.5 (6.97)	57.9 (9.53)	0.880
HR_MAX_ (bpm)	184.9 (11.11)	183.2 (8.68)	0.593
VT1 (%)	50.9 (5.63)	50.0 (4.78)	0.610
VT2 (%)	84.9 (5.85)	84.1 (5.70)	0.644

Values are expressed as mean (SD). BMI = body mass index; BF = body fat; VO2máx = maximum oxygen volume; VT1 = ventilatory threshold 1 (aerobic); VT2 = ventilatory threshold 2 (anaerobic).

**Table 2 biology-11-00736-t002:** Training variables of the cyclists.

	2S-Hesperidin	Placebo	
Total distance (km)	1121.12 (534.99)	1082.43 (810.46)	0.868
HR_AVG_ (bpm)	144.76 (8.88)	137.48 (13.11)	0.067
W_AVG_ (W)	174.86 (15.79)	163.47 (32.49)	0.435
RPE	6.34 (0.82)	6.33 (1.16)	0.975

Values are expressed as mean (SD). Total distance = of all the training sessions carried out during the study period; HRavg = average heart rate of all the training sessions carried out during the study period; Wavg = average power output of all training sessions during the study period and RPE = rating of perceived exertion of all training sessions during the study.

**Table 3 biology-11-00736-t003:** Between-group comparisons in dietary intake of cyclists.

	Pre-Intervention			Post-Intervention		
	2S-Hesperidin	Placebo	*p*-Value	2S-Hesperidin	Placebo	*p*-Value
Kcal	2163.6 (519.02)	2100.2 (515.77)	0.708	1974.1 (377.97)	2133.5 (437.98)	0.237
Kcal/BM	31.1 (9.34)	30.2 (8.71)	0.768	27.9 (6.53)	30.3 (6.46)	0.249
CHO (g)	245.7 (73.46)	222.0 (69.68)	0.312	216.6 (63.47)	248.3 (58.15)	0.117
CHO/Kg b.w.	3.5 (1.31)	3.2 (1.14)	0.416	3.1 (1.08)	3.5 (0.94)	0.173
PRO (g)	113.5 (25.21)	115.2 (25.37)	0.837	109.0 (23.05)	101.5 (23.67)	0.332
PRO/Kg b.w.	1.6 (0.41)	1.7 (0.48)	0.778	1.5 (0.35)	1.5 (0.42)	0.596
LP (g)	80.8 (27.24)	83.5 (23.65)	0.739	71.5 (17.61)	71.6 (18.89)	0.985
LP/Kg b.w.	1.2 (0.45)	1.2 (0.37)	0.758	1.0 (0.27)	1.0 (0.29)	0.823

Values are expressed as mean (SD). Kcal = kilocalories; CHO = carbohydrates; PRO = protein; LP = lipids; BM = body mass. The mean values correspond to the average of all 24-h diet recall data collected at pre-intervention (visits twoand three) and post-intervention (visits four and five).

**Table 4 biology-11-00736-t004:** Changes in the acid-base state and metabolic substrate in capillary finger blood at baseline, FatMax1, ventilatory threshold 1 (VT1), ventilatory threshold 2 (VT2), power output maximum (P_MAX_), FatMax2 and EPOC during the rectangular test. Values are mean (SD) and effect size (ES).

		2S-Hesperidin								Placebo							
		Baseline	FatMax1	VT1	VT2	Pmax	FatMax2	EPOC	AUC	Baseline	FatMax1	VT1	VT2	Pmax	FatMax2	EPOC	AUC
pH	Pre	7.417 (0.01)	7.395 (0.01)	7.392 (0.01)	7.326 (0.01)	7.202 (0.02)	7.344 (0.01)	7.395 (0.01)	44.07 (0.13)	7.422 (0.01)	7.398 (0.01)	7.405 (0.01)	7.320 (0.02)	7.188 (0.02)	7.341 (0.01)	7.399 (0.01)	44.07 (0.17)
	Post	7.404 (0.01)	7.419 (0.02)	7.402 (0.01)	7.342 (0.01)	7.213 (0.01)	7.363 (0.01)	7.403 (0.01)	44.14 (0.08)	7.407 (0.01)	7.388 (0.02)	7.399 (0.01)	7.328 (0.01)	7.202 (0.02)	7.352 (0.01)	7.394 (0.01)	44.07 (0.15)
	*p*-value	0.248	0.131	0.121	0.254	0.364	**0.028**	0.213	**0.016**	0.238	0.564	0.317	0.572	0.292	0.231	0.412	0.895
	ES	1.11	3.21	1.87	1.07	0.69	1.48	1.25	0.54	1.15	1.15	1.11	0.46	0.79	0.78	0.77	0.00
Bicarbonate anion (mmol/L) (HCO_3_^−^)	Pre	27.09 (0.73)	25.63 (0.51)	25.03 (0.41)	20.32 (1.07)	13.98 (0.62)	19.43 (0.90)	24.29 (0.50)	130.72 (10.40)	25.76 (0.79)	24.92 (0.55)	25.49 (0.44)	18.43 (1.17)	13.01 (0.68)	18.64 (0.98)	23.50 (0.54)	125.00 (12.27)
	Post	26.85 (0.32)	28.27 (1.18)	26.42 (0.34)	21.65 (0.55)	14.71 (0.63)	21.53 (0.76)	25.69 (0.42)	139.01 (8.94)	25.94 (0.35)	25.15 (1.29)	25.06 (0.37)	20.04 (0.60)	13.94 (0.69)	19.36 (0.83)	24.30 (0.46)	128.54
	*p*-value	0.744	**0.040**	**<0.001**	0.200	0.204	**0.003**	**0.001**	**0.012**	0.828	0.864	0.231	0.156	0.136	0.304	**0.045**	0.122
	ES	0.31	4.85	3.20	1.16	1.08	2.19	2.63	0.74	0.20	0.38	0.89	1.27	1.26	0.69	1.36	0.26
Standard bicarbonate (mmol/L) (SBC)	Pre	26.29 (0.76)	24.79 (0.46)	24.31 (0.29)	20.19 (0.84)	14.67 (0.49)	20.04 (0.72)	23.88 (0.42)	129.73 (7.73)	25.41 (0.86)	24.36 (0.52)	24.97 (0.34)	18.85 (0.96)	13.94 (0.56)	19.59 (0.82)	23.56 (0.47)	126.23 (9.58)
	Post	25.78 (0.24)	27.36 (1.35)	25.44 (0.31)	21.16 (0.44)	15.22 (0.49)	21.58 (0.61)	24.95 (0.33)	136.31 (7.67)	25.12 (0.28)	24.28 (1.54)	24.61 (0.35)	19.74 (0.50)	14.76 (0.56)	20.31 (0.70)	24.03 (0.37)	128.30 (8.21)
	*p*-value	0.516	**0.076**	**0.001**	0.222	0.165	**0.006**	**0.001**	**0.017**	0.744	0.960	0.267	0.322	**0.072**	0.227	0.164	0.111
	ES	0.63	5.26	3.59	1.08	1.04	2.01	2.39	0.79	0.31	0.14	0.98	0.85	1.34	0.80	0.90	0.20
Lactate (mmol/L) (Lac)	Pre	1.66 (0.14)	2.62 (0.32)	2.53 (0.24)	8.45 (0.88)	14.46 (0.86)	6.98 (0.74)	3.61 (0.42)	36.72 (8.50)	1.77 (0.15)	2.92 (0.34)	2.25 (0.27)	8.68 (0.96)	15.35 (0.93)	7.46 (0.81)	4.23 (0.45)	39.32 (6.76)
	Post	1.68 (0.13)	1.85 (0.24)	1.75 (0.23)	7.05 (0.52)	14.21 (0.79)	5.49 (0.67)	2.94 (0.35)	32.10 (6.58)	1.84 (0.15)	2.28 (0.26)	2.30 (0.25)	8.35 (0.56)	14.82 (0.86)	7.22 (0.73)	3.65 (0.38)	37.47 (8.54)
	*p*-value	0.871	**0.010**	**0.003**	0.134	0.730	**0.018**	**0.039**	**0.057**	0.680	**0.041**	0.833	0.741	0.503	0.702	0.098	0.391
	ES	0.15	2.26	2.98	1.49	0.28	1.88	1.51	0.51	0.38	1.71	0.19	0.32	0.53	0.28	1.17	0.25
Glucose (mg/dL) (Glu)	Pre	100.00 (2.91)	86.25 (2.97)	93.58 (3.28)	96.83 (3.37)	129.08 (6.17)	114.25 (6.08)	101.00 (6.45)	616.46 (68.76)	101.82 (3.04)	87.00 (3.10)	87.82 (3.43)	94.27 (3.52)	123.73 (6.44)	112.73 (6.35)	93.00 (6.74)	611.50 (73.23)
	Post	100.00 (2.57)	87.75 (3.24)	95.42 (2.79)	95.08 (3.58)	118.67 (5.92)	109.83 (5.38)	99.58 (3.62)	598.91 (69.45)	98.55 (2.69)	91.09 (3.38)	89.00 (2.91)	93.82 (3.74)	116.91 (6.19)	106.73 (5.62)	92.73 (3.78)	591.40 (64.12)
	*p*-value	1.000	0.677	0.589	0.616	0.090	0.487	0.782	0.248	0.231	0.282	0.738	0.900	0.277	0.368	0.959	0.255
	ES	0.00	0.47	0.52	0.48	1.57	0.68	0.20	0.24	0.99	1.22	0.32	0.12	0.98	0.87	0.04	0.25
Actual base excess (mmol/L) (ABE)	Pre	2.14 (0.79)	0.50 (0.52)	−0.02 (0.35)	−5.21 (1.14)	−13.13 (0.78)	−5.20 (1.03)	−0.52 (0.50)	26.57 (8.72)	1.23 (0.90)	0.18 (0.59)	0.75 (0.40)	−7.02 (1.30)	−14.25 (0.89)	−6.09 (1.17)	−0.91 (0.57)	30.22 (10.04)
	Post	1.72 (0.28)	3.17 (1.31)	1.28 (0.36)	−3.84 (0.57)	−12.28 (0.74)	−3.42 (0.79)	0.76 (0.38)	24.13 (8.72)	0.92 (0.32)	−0.12 (1.49)	0.30 (0.41)	−5.71 (0.65)	−13.00 (0.85)	−5.12 (0.90)	−0.35 (0.44)	26.76 (8.15)
	*p*-value	0.604	**0.059**	**0.001**	0.200	0.169	**0.034**	**0.001**	0.472	0.739	0.846	0.238	0.279	0.082	0.290	0.154	0.103
	ES	0.50	4.82	3.48	1.13	1.02	1.62	2.42	0.26	0.31	0.46	1.04	0.92	1.28	0.76	0.90	0.32
Standard base excess (mmol/L) (SBE)	Pre	2.59 (0.85)	0.72 (0.59)	0.08 (0.42)	−5.67 (1.25)	−14.04 (0.81)	−5.93 (1.43)	−0.62 (0.57)	29.22 (9.82)	1.31 (0.92)	0.06 (0.64)	0.79 (0.45)	−7.67 (1.36)	−15.25 (0.88)	−5.59 (1.56)	−1.31 (0.62)	33.13 (10.89)
	Post	2.14 (0.33)	3.77 (1.41)	1.61 (0.41)	−4.10 (0.65)	−10.71 (1.79)	−3.88 (0.90)	0.92 (0.46)	26.76 (9.37)	1.21 (0.36)	0.15 (1.54)	0.51 (0.45)	−5.94 (0.71)	−14.08 (1.95)	−6.24 (0.97)	−0.62 (0.50)	29.67 (8.62)
	*p*-value	0.606	**0.046**	**0.001**	0.187	0.106	0.128	**<0.001**	0.509	0.916	0.954	0.500	0.180	0.591	0.651	0.104	0.174
	ES	0.50	4.87	3.42	1.17	3.87	1.34	2.53	0.23	0.10	0.13	0.57	1.18	1.23	0.38	1.03	0.29

Pre = before the eight-week intervention with 2S-hesperidin and placebo, post = after the eight-week intervention with 2S-hesperidin and placebo. In bold are *p*-values = ≤ 0.05 and trends between 0.05–0.08.

**Table 5 biology-11-00736-t005:** Comparison of pre-post-intervention differences between each of the rectangular test points (Baseline, FatMax1, VT1, VT2, P_MAX_, FatMax2, EPOC) and AUC between groups. Values are mean (SD).

		Between-Group Comparison							
		ΔBaseline	ΔFatMax1	ΔVT1	ΔVT2	ΔP_MAX_	ΔFatMax2	ΔEPOC	ΔAUC
pH	Differences	−0.029 (0.23)	−0.072 (0.32)	0.198 (0.28)	−0.149 (0.27)	−0.136 (0.36)	0.055 (0.24)	0.066 (0.28)	0.077 (0.03)
	*p*-value	0.90	0.82	0.49	0.58	0.71	0.83	0.81	**0.02**
	Effect size	0.17	2.13	2.50	0.52	0.24	0.94	2.01	1.03
HCO_3_^−^ (mmol/L)	Differences	−2.650 (5.33)	−4.495 (4.14)	−4.053 (3.90)	10.906 (7.43)	2.786 (19.73)	2.013 (3.34)	5.558 (6.17)	4.752 (3.57)
	*p*-value	0.62	0.29	0.31	0.18	0.89	0.55	0.37	0.20
	Effect size	0.55	1.92	5.48	0.27	0.36	2.08	1.66	0.59
SBC (mmol/L)	Differences	−0.100 (0.16)	0.348 (0.19)	−0.063 (0.26)	0.079 (0.47)	0.777 (0.53)	0.035 (0.29)	0.134 (0.17)	4.522 (2.63)
	*p*-value	0.55	**0.07**	0.81	0.87	0.15	0.91	0.44	0.11
	Effect size	0.27	1.81	5.07	0.10	0.68	1.53	1.95	0.67
Lac (mmol/L)	Differences	0.060 (0.70)	−0.118 (0.60)	0.974 (0.71)	−0.760 (0.96)	1.055 (0.84)	0.372 (0.43)	0.484 (0.80)	−2.772 (3.03)
	*p*-value	0.93	0.85	0.18	0.43	0.22	0.40	0.55	0.37
	Effect size	0.28	0.45	3.39	1.15	0.37	2.04	0.30	0.39
Glu (mg/dL)	Differences	0.635 (0.74)	0.435 (0.44)	−0.511 (0.64)	0.918 (0.93)	−0.872 (0.79)	−0.096 (0.32)	0.205 (0.62)	2.558 (21.82)
	*p*-value	0.40	0.33	0.43	0.33	0.28	0.77	0.74	0.91
	Effect size	1.26	0.71	0.19	0.37	0.60	0.25	0.22	0.18
ABE (mmol/L)	Differences	3.550 (3.23)	0.853 (4.13)	7.737 (4.30)	6.515 (3.91)	−2.786 (7.51)	2.524 (6.50)	0.211 (5.42)	1.020 (4.00)
	*p*-value	0.28	0.84	0.08	0.11	0.71	0.70	0.97	0.80
	Effect size	0.13	2.09	5.05	0.05	0.62	0.97	2.05	0.10
SBE (mmol/L)	Differences	0.763 (3.65)	−1.609 (3.42)	−8.400 (2.86)	−5.592 (3.81)	1.642 (8.82)	−6.097 (5.26)	−4.703 (3.58)	0.994 (4.50)
	*p*-value	0.84	0.64	**0.01**	0.15	0.85	0.25	0.20	0.83
	Effect size	0.39	1.98	4.59	0.14	1.05	2.00	2.17	0.07

HCO_3_^−^ = bicarbonate anion, SBC = standard bicarbonate, Lac = lactate, Glu = glucose, ABE = actual base excess and SBE = standard base excess. In bold are *p*-values = ≤ 0.05 and trends between 0.05–0.08.

## Data Availability

The data presented in this study are available on request from the corresponding author. The data are not publicly available due to privacy reasons.

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
