# Peer review of "Chronic Supplementation of 2S-Hesperidin Improves Acid-Base Status and Decreases Lactate at FatMax, at Ventilatory Threshold 1 and 2 and after an Incremental Test in Amateur Cyclists"

_biology, 2022, doi:10.3390/biology11050736_

Round 1

Reviewer 1 Report

Comments:

  1. Title: It is advised to use suitable word instead of ‘8-week’ or simply can present ‘supplementation of…..’. It is unclear from “at Low-moderate and Sub-maximal Intensities and After an Incremental Test in Amateur Cyclists”. Please precise the title for clarity.
  2. Abstract: Blood collection time points were not specific.
  3. Although there is a word limit in the Abstract, excessive and unnecessary abreactions in the Abstract are confusing. Please provide the level of significance (p values) for important data.
  4. Line 90-94: It seems author hypothesized something.. but used previous studies as citations. Please check it.
  5. Methods: Please provide the nutritional composition of 2S-hesperidin.
  6. What was the time of blood collection?
  7. Please provide the suitable sub-titles for relevant biomarkers under Results section, and explain their results accordingly.

Author Response

We thank the reviewer for their constructive and helpful feedback on our manuscript. We have replied to each specific comment in the section below and have introduced the corresponding edits into the manuscript using Word’s track changes.

Title: It is advised to use suitable word instead of ‘8-week’ or simply can present ‘supplementation of…..’. It is unclear from “at Low-moderate and Sub-maximal Intensities and After an Incremental Test in Amateur Cyclists”. Please precise the title for clarity.

Response. Following your suggestion, we have changed the title to make the title clearer.

Abstract: Blood collection time points were not specific.

Response: The abstract specifies the blood draws at the time they are performed (baseline, FatMax1, VT1, VT2, PMAX, FatMax2 and EPOC). In the methodology, the blood collection points in the rectangular test are also described in detail.

Although there is a word limit in the Abstract, excessive and unnecessary abreactions in the Abstract are confusing. Please provide the level of significance (p values) for important data.

Response: The abbreviations are common in this area and writing them out would exceed the word limit. Following your suggestion, we have entered the statistical data in the abstract.

Line 90-94: It seems author hypothesized something. but used previous studies as citations. Please check it.

Response: We have removed the citations in the hypothesis and introduced the papers in lines 

Methods: Please provide the nutritional composition of 2S-hesperidin.

Response: Following your suggestion, we have introduced the nutritional composition of the Cardiose® capsules. Lines 127-132.

What was the time of blood collection?

Response: There were 2 types of blood draws, those in point 2.3.5. of the methodology which were blood draws from the anterocubital vein, to find out if the subjects were healthy at visit 1. And in point 2.3.6. there were the capillary blood draws from the finger which were performed in the rectangular test at visits 3 and 5. To avoid confusion, we have moved point 2.3.5. to 2.3.2.

Please provide the suitable sub-titles for relevant biomarkers under Results section, and explain their results accordingly.

Response: Amended.

Author comment: We appreciate all the comments made on our manuscript, which helped improve it’s quality.

Reviewer 2 Report

The manuscript is part of a wider project, but too often the authors refer to their previously published data, the manuscript should be able to stand alone.

The introduction is fine and it presents the topic.

The methodology must be improved. The study design description can be improved, how different biomarkers were analyzed must be detailed, how did the authors measure Lactate, bicarbonate, pH, Abe SBE, and all other parameters? what is their biological meaning?

Line 106-107: please detail which baseline blood analyses were considered.

Table 1: I do not think Training variables can be included in baseline parameters, because they refer to the study period (Hesperidig group took the supplement in this period so it is not baseline).

Figure 1: divide fig 1 into fig 1A (study plan) and fig 1B rectangular test. in figure 1A include a timeline (in days or weeks) moreover if you draw the pill symbol only at V2 and V5 it seems the subject took the supplement only at that two points and not from V2 to V5.

In the rectangular test figure: how did you set FatMax, VT1, VT2, are they a specific % of VO2max?

Table 2: don't you think a 2100 Kcal/day intake is a very low intake for people performing 6-12 h/wk training? can you better explain how you calculate energy and nutrient intake from FFQ?

Table 2: /BM should be converted into /Kg b.w.

Line 202: (pre) call it baseline, otherwise, in table 3 you will have 2 pre.

Table 3: does "Post" mean after 8 weeks of supplementation? it is not clear.

Table 3: data are expressed as mean (SE) but in the statistics section, you wrote standard dev.

Table 3: ES line, is it effect size? you should mention it in the caption.

Table 4 is absolutely not clear. How did you calculate these differences between groups? Is it delta baseline for Hesperidin group - delta baseline for placebo? Moreover, in the results section these data are not presented. What do these differences represent and mean?

Lines 252-258 paragraph 4.1 is not clear, why are you talking about the sept-dec period? has it ever been mentioned before?

Lines 256-258 sentence is not clear.

The discussion is quite long and more important it does not provide any explanation for the Hesperidin effect. how did it improve acid-base balance remains unknown.

Author Response

We thank the reviewer for their constructive and helpful feedback on our manuscript. We have replied to each specific comment in the section below and have introduced the corresponding edits into the manuscript using Word’s track changes.

The manuscript is part of a wider project, but too often the authors refer to their previously published data, the manuscript should be able to stand alone.

Response: We understand the reviewer’s concern. However, because few studies have evaluated the effect of 2S-hesperidin intake on sports performance, we felt it necessary to incorporate our published studies and those of others (few in animals and humans) to generate a reliable background that encompasses all the evidence on this subject, without wanting to convey any bad intentions.

The introduction is fine and it presents the topic.

Response. Thanks.

The methodology must be improved. The study design description can be improved, how different biomarkers were analyzed must be detailed, how did the authors measure Lactate, bicarbonate, pH, Abe SBE, and all other parameters? what is their biological meaning?

Response: We have modified the methodology description based on Reviewer 1’s comments. In addition, we have clarified the different biomarkers anayzled with capillary blood (lines 195-198).

Line 106-107: please detail which baseline blood analyses were considered.

Response: Amended. Line 111.

Table 1: I do not think Training variables can be included in baseline parameters, because they refer to the study period (Hesperidig group took the supplement in this period so it is not baseline).

Response: Following your suggestion, we have separated the training variables into the new Table 2 from the other variables in Table 1.

Figure 1: divide fig 1 into fig 1A (study plan) and fig 1B rectangular test. in figure 1A include a timeline (in days or weeks) moreover if you draw the pill symbol only at V2 and V5 it seems the subject took the supplement only at that two points and not from V2 to V5.

Response: Following your suggestion, we have modified figure 1. We have clarified the duration of supplementation in the legend.

In the rectangular test figure: how did you set FatMax, VT1, VT2, are they a specific % of VO2max?

Response: No, it was defined by Wasserman (ref 41).

Table 2: don't you think a 2100 Kcal/day intake is a very low intake for people performing 6-12 h/wk training? can you better explain how you calculate energy and nutrient intake from FFQ?

Response: At a nutritional level, no modifications were made. We only collected their eating habits. We believe that the eating intake was not very high for the weekly hours of training, but as they were in a period of lower volume and intensity, it is possible that they had reduced their calorie and carbohydrate intake.

Table 2: /BM should be converted into /Kg b.w.

Response: Amended. 

Line 202: (pre) call it baseline, otherwise, in table 3 you will have 2 pre.

Response: Amended. 

Table 3: does "Post" mean after 8 weeks of supplementation? it is not clear.

Response: To clarify the abbreviation "Post" we have defined it in the legend.

Table 3: data are expressed as mean (SE) but in the statistics section, you wrote standard dev.

Response: Table 3, which is now table 4, shows the data as mean (standard deviation; SD). Thank you for catching this error.

Table 3: ES line, is it effect size? you should mention it in the caption.

Response: That's right, ES represents the effect size. Amended

Table 4 is absolutely not clear. How did you calculate these differences between groups? Is it delta baseline for Hesperidin group - delta baseline for placebo? Moreover, in the results section these data are not presented. What do these differences represent and mean?

Response: Table 4, which is now the Table 5., indicates between-group comparison of pre-post-intervention differences between each of the rectangular test points (Baseline, FatMax1, VT1, VT2, PMAX, FatMax2, EPOC) and AUC. In the results section, this data appears with the symbol ∆, for example in lines 238 and 261.

Lines 252-258 paragraph 4.1 is not clear, why are you talking about the sept-dec period? has it ever been mentioned before?

Response: This paragraph was introduced to make it known that this study was carried out at a time when cyclists decreased the volume and intensity of training and that some studies have shown changes at a physiological biochemical level. Depending on the time of the sporting season when the studies are carried out, especially those with chronic effects, adaptations to training can generate biases when evaluating supplements.

Lines 256-258 sentence is not clear.

Response: Other studies have shown a decrease in the adaptation in the variables of interest. However, we wanted to acknowledge that our study did not show such result.

The discussion is quite long and more important it does not provide any explanation for the Hesperidin effect. how did it improve acid-base balance remains unknown.

Response: We understand your concern, but to our knowledge, no studies have evaluated the effect of 2S-hesperidin on acid-base balance in either animals or humans. We are therefore unable to develop a possible hypothesis in this case. Future studies are needed to investigate possible molecular pathways that could modulate 2S-hesperidin intake in humans.

Author comment: We appreciate all the comments made on our manuscript, which helped improve it’s quality.

Round 2

Reviewer 2 Report

All my questions have been replied properly, I feel the manuscript is not ready for publication.